# A New Approach Determining a Phase Transition Boundary Strictly Following a Definition of Phase Equilibrium: An Example of the Post-Spinel Transition in Mg₂SiO₄ System

**Takayuki Ishii [1,\*], Artem Chanyshev [2] and Tomoo Katsura [2]**

[1] Center of High Pressure Science and Technology Advanced Research (HPSTAR), Beijing 100094, China

[2] Bayerisches Geoinstitut, University of Bayreuth, 95440 Bayreuth, Germany; artem.chanyshev@uni-bayreuth.de (A.C.); tomo.katsura@uni-bayreuth.de (T.K.)

\* Correspondence: takayuki.ishii@hpstar.ac.cn

**Abstract:** The Clapeyron slope is the slope of a phase boundary in P–T space and is essential for understanding mantle dynamics and evolution. The phase boundary is delineating instead of balancing a phase transition's normal and reverse reactions. Many previous high pressure–temperature experiments determining the phase boundaries of major mantle minerals experienced severe problems due to instantaneous pressure increase by thermal pressure, pressure drop during heating, and sluggish transition kinetics. These complex pressure changes underestimate the transition pressure, while the sluggish kinetics require excess pressures to initiate or proceed with the transition, misinterpreting the phase stability and preventing tight bracketing of the phase boundary. Our recent study developed a novel approach to strictly determine phase stability based on the phase equilibrium definition. Here, we explain the details of this technique, using the post-spinel transition in Mg₂SiO₄ determined by our recent work as an example. An essential technique is to observe the change in X-ray diffraction intensity between ringwoodite and bridgmanite + periclase during the spontaneous pressure drop at a constant temperature and press load with the coexistence of both phases. This observation removes the complicated pressure change upon heating and kinetic problem, providing an accurate and precise phase boundary.

**Keywords:** phase equilibrium; phase transition; Clapeyron slope; phase transition kinetics

## 1. Introduction

The Clapeyron slope, the temperature dependence of a phase transition pressure, of major constituents in mantle rocks is an essential thermodynamic property for understanding mantle dynamics. If the Clapeyron slope is positive and negative, respectively, the phase transition enhances and impedes mantle convection [1]. Many geodynamicists simulated mantle convections by considering the Clapeyron slopes to demonstrate their impact on mantle dynamics and evolution [2–10].

The Clapeyron slopes of major mantle phase transitions, such as the olivine–wadsleyite–ringwoodite transitions and dissociation of ringwoodite to bridgmanite + periclase, have been experimentally determined in multi-anvil apparatus [11–13]. Its combination with in situ X-ray diffraction can determine the Clapeyron slopes most accurately and precisely among various experimental and theoretical approaches [14]. One reason is the highly stable and homogenous temperature field produced by the resistively heated furnace, allowing a temperature resolution of 3 K at temperatures around 2000 K [15–17]. A second reason is the precise pressure determination by measuring relative volumes of pressure-standard materials by using X-ray diffraction. The most advanced technology allows a pressure resolution of 0.05 GPa in the pressure range to 30 GPa [15–17]. A third is the nearly real-time observation of changes in phase assemblies and their proportions. These

advantages enable determining Clapeyron slopes robustly to discuss mantle structure and dynamics in detail based on these slopes [15–27].

Although this technique has provided us with the most robust results, there are discrepancies among previous studies due to various problems that occur during heating. To solve these problems, Chanyshev et al. [17] developed a novel method using a multi-anvil press to determine a phase boundary strictly based on the definition of phase equilibrium and obtained the boundaries of the dissociation of ringwoodite to bridgmanite + periclase (post-spinel transition) in $Mg_2SiO_4$ and akimotoite–bridgmanite transition in $MgSiO_3$. In this paper, we explain the essential points of this technique in detail by taking the post-spinel transition in $Mg_2SiO_4$ as an example. We note that the general experimental method for in situ X-ray diffraction in a multi-anvil press for this experiment was described in [17].

## 2. Definition of Phase Equilibrium

We first recall the principle of phase equilibrium. Since we are interested in high-pressure–temperature phase transitions, we consider in this argument a phase transition between one phase that is stable at higher pressures, referred to as the H.P.-phase, and another phase that is stable at lower pressures, referred to as the L.P.-phase (Figure 1). Note that the L.P.-phase is unstable at ambient pressure. In principle, the phase transitions from the L.P.-phase to the H.P.-phase and from the H.P.-phase to the L.P.-phase should occur under any conditions. The former and latter, respectively, are referred to as the normal and reversal reactions. The "stability field of the H.P.-phase" means that the region where the transition rate from the L.P.-phase to the H.P.-phase is higher than that from the H.P.-phase to the L.P.-phase, and vice versa. The definition of phase equilibrium is the balance of these two transitions. The "phase boundary" is the trajectory of conditions of phase equilibrium. In many high-pressure experiments, a formation of either phase from a starting material, whose stability field is located even lower than that of the L.P.-phase, has been regarded as evidence of its stability. However, forming a new phase from a starting material cannot be such evidence but means that the newly formed phase is more stable than the starting material. Therefore, we need to observe the formation of either phase from the other.

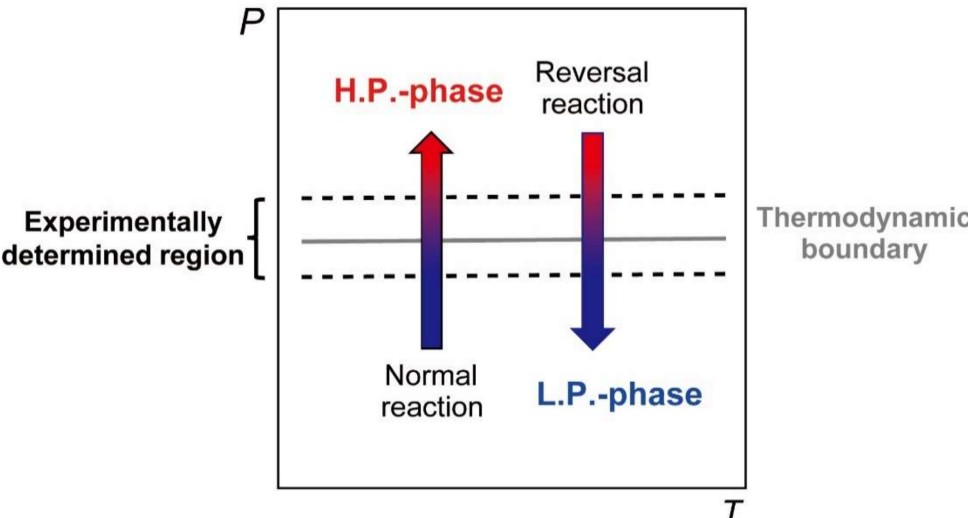

**Figure 1.** Determination of a phase transition pressure between low pressure (L.P.)- and high pressure (H.P.)-phases based on the principle of phase equilibrium.

Note that it is impractical to find the conditions of the equal transition rates because "no change" should be observed under such conditions. Commonly, sluggish kinetics prevents the progress of reactions near the phase boundary. Moreover, the surface energy prevents the formation of new grains of a stable phase. As a result, there is a region with a finite width where neither reaction proceeds. Such ranges are often too broad

for geophysical application [23]. In some studies, the observation of granular texture is regarded as evidence of stability [12]. This argument is also incorrect because a metastable phase can survive for a long time to show granular texture.

The practical determination of a phase boundary is to let the regions where the normal and reversal reactions proceed bracket a band (Figure 1). In this aspect, observing changes in the relative amounts of the H.P.- and L.P.-phases in real-time using in situ X-ray diffraction is essential. Minimizing the kinetic effect is needed to narrow the bracketed region for geophysical application.

## 3. Problems with Previous Experiments

This section discusses problems with previous studies combining the multi-anvil experiment and in situ X-ray observation. These problems may have hindered the accurate and precise determination of phase boundaries due to the samples' complicated environment and property changes during heating.

One such potential problem is the pressure drop. Ishii et al. [15,16] showed that sample pressure in a multi-anvil press first rapidly and then slowly drops after reaching the target temperature (Figure 2). The pressure drop in a multi-anvil press happens for various reasons, such as the release of deviatoric stresses stored in a sample during cold compression, material softening, sample sintering, and sample phase transitions. The material softening includes those of second-stage anvils, gaskets, and pressure media. The drop rate gradually becomes smaller with time.

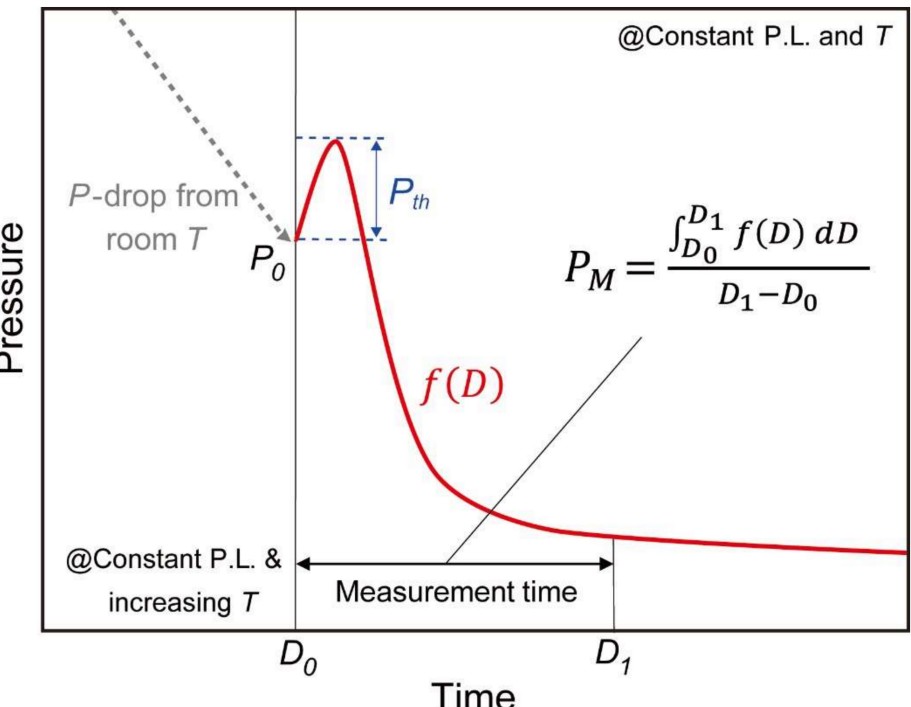

**Figure 2.** A schematic drawing of pressure drop during and after heating after cold compression at a fixed press load (P.L.). During heating, sample pressure significantly decreases mainly due to stress relaxation. After reaching the target temperature, an initial pressure ($P_0$) first rapidly increases in a few seconds due to thermal pressure ($P_{th}$) and decreases later, and then slowly decreases. Sample pressure ($P_M$) between $D_0$ and $D_1$ is an average pressure during the measurement.

Another potential problem is the pressure increase by the thermal pressures of the sample and pressure media. This pressure increase should happen instantaneously, although our experimental technique cannot observe it due to the subsequent pressure drop mentioned above and a much longer time (order of a minute) of the pressure measurement

than the time-scale of initial pressure change (order of a second) (Figure 2). If this temperature increase were too large, the sample could experience higher pressure than indicated by in situ X-ray diffraction, causing the phase transition [23]. Since phase transitions could already be completed when taking the first sample diffraction, the transition pressures should be uncertain in such circumstances. Since the magnitude of the sum of the thermal pressure and initial pressure drop upon heating should be changed by the target temperature and heating rate, previous studies could have misinterpreted the Clapeyron slopes.

We also think that the accuracy in the reported temperature of phase transitions is doubtful because the reaction-start temperature could be lower than the target temperature in most cases. Reconstructive phase transitions, which are the case with many mantle phase transitions, do not necessarily occur upon crossing phase boundaries in heating at relatively low temperatures and would require significant excess temperature due to sluggish kinetics. Phase transitions start at reaction kinetics' threshold temperatures and could complete before reaching the target temperatures if the target temperatures are higher than the threshold temperatures. In these circumstances, phase transitions could have happened at lower temperatures than the reported target temperatures, misinterpreting transition temperatures.

One may consider that these problems associated with the sluggish kinetics should be solved or at least suppressed at higher temperatures because the reaction kinetics is expected to follow the Arrhenius relation. Unfortunately, this expectation is not always realized because the defect structures and populations should affect the reaction kinetics in solid–solid transitions. Powdered starting materials have spaces among grains before compression, and filling these spaces causes intense local stresses during cold compression, producing high-density defects and making the starting materials highly reactive. Starting materials can quickly transform into high-pressure phases for this reason. However, once they transform to high-pressure phases, their point- and line-defect densities are low, making the newly formed high-pressure phases inert. If the high-pressure phases form at relatively low temperatures, the grain sizes are small, and the phase transition can occur on the grain boundaries. However, with increasing temperature, grain growth occurs, decreasing the grain-boundary areas and making the high-pressure phase more inert. Since the grain growth proceeds with time, samples become more inert while staying at high temperatures for longer durations. Therefore, the stepwise increase in sample temperature with taking diffractions of the samples and pressure standards, identifying phases present and calculating pressures, which were adopted in many past investigations, hinders phase transitions. The production of large, high-crystallinity grains may be why the post-spinel transition and the akimotoite–bridgmanite transition were challenging to initiate at higher temperatures in earlier studies [22,23,26,28].

Below, we more concretely explain the procedure adopted in previous studies and the impact of the abovementioned problems on the results, in the case of the negative Clapeyron slope. After cold compression, a starting material is heated to a target temperature and becomes the H.P.- or L.P.-phase depending on the pressure–temperature path. If the starting material were brought above the phase boundary, it would become the H.P.-phase. If the target temperature were higher than the threshold temperature, it would become the H.P.-phase before reaching the target temperature, overestimating the transition temperature. After reaching the target temperature, the H.P.-phase that already appeared might persist due to its sluggish kinetics even when sample pressure decreased below the phase boundary by pressure drop, overestimating the transition pressure (Figure 3a). On the other hand, if the sample passed below the phase boundary at the threshold temperature, the starting sample would first become the L.P.-phase. If the sample crossed the phase boundary before reaching a target temperature, the L.P.-phase might persist even above the boundary due to sluggish kinetics, overestimating the transition pressure (Figure 3b). After the synthesis, the pressure–temperature condition of the sample would be changed to bring the sample to the opposite side of the phase boundary (Figure 3c). However, excess pressure should be needed to start the transformation due to the sluggish kinetics. Thus, the phase stability

can be misinterpreted when the kinetic effect is not considered. Even if the kinetic effect were considered, the phase stability could not be constrained tightly due to the sluggish kinetics near the boundary.

**Figure 3.** A schematic drawing of problems on in situ determination in a multi-anvil press. (**a**) After cold compression, when a starting material (S.M.) was brought above the phase boundary, it could become the H.P.-phase during heating if the target temperature were higher than the threshold temperature of the transition. After reaching the target temperature, the H.P.-phase might have already appeared and persist due to its sluggish kinetics even though the sample was below the phase boundary by pressure drop. (**b**) When the sample passed below the phase boundary during heating, the starting sample should first become the L.P.-phase at the threshold temperature. If the sample crossed the phase boundary until it reached a target temperature, the L.P.-phase might persist even above the boundary due to the sluggish kinetics. (**c**) When the in situ synthesized sample crossed the phase boundary, excess pressure should be needed to initiate the transition due to fewer defects in the structure. Thus, the phase boundary could be either underestimated or overestimated.

By considering these circumstances, a novel strategy is necessary to determine transition pressure and temperature accurately.

## 4. A Novel Strategy to Determine a Phase Boundary

To obtain an accurate and precise phase boundary between two phases based on the definition of phase equilibrium explained in Section 2, and by overcoming the problems explained in Section 3, we consider the following procedure (Figure 4). The following numbered items correspond to the numbers shown in Figure 4.

(1) The starting material is the H.P.-phase pre-synthesized at the lowest temperature, where the H.P.-phase can be synthesized. Since considerable excess Gibbs energy can drive the phase transition to form the H.P.-phase by overcoming the sluggish kinetics resulting from the low temperature, the synthesis pressure should be as high as possible.

(2) We load the starting material of the H.P.-phase in a multi-anvil cell, compress it, and heat it to the lowest possible temperature where the H.P.-phase can transform to the L.P.-phase to convert the H.P.-phase partially to the L.P.-phase and obtain the coexistence of the H.P.- and L.P.-phases. The pressure should be the lowest in the L.P.-phase stability field to realize the maximum deficient Gibbs energy and enables the partial transition of the H.P.-phase to the L.P.-phase at the lowest temperature. However, since the lower limit of the L.P.-phase stability field is often not well determined, the pressure is higher in practice. This procedure also releases the sample stresses

stored during the initial compression, which helps decrease the pressure drop in the later stages. In many cases, the heating in this step reduces the sample pressure by several GPa.

(3) After a sufficient amount of L.P.-phase is obtained to observe using X-ray diffraction and to trace the relative intensity change in the phases, we cool the sample by several hundred K to stop the transition and bring it to higher pressures than the supposed phase boundary.

(4) We heat the sample to the lowest temperature, where the phase transitions can occur kinetically.

(5) We take diffraction patterns of the pressure standard and sample alternatively by keeping the press load and temperature constant. The pressure spontaneously drops gradually. We measure the sample pressure and observe the change in the ratio of the H.P.-phase to the L.P.-phase from the diffraction intensity at constant temperature and monotonically decreasing pressure. L.P.-phase should slowly transform into the H.P.-phase at the beginning because the sample is initially in the stability field of the H.P.-phase, as written in (3). If the pressure drop ceases, we slightly decrease the press load to enhance the pressure drop. We note that the sample pressure is immediately changed by the decrease in press load and kept constant because factors for complicated pressure change shown in Figure 2 have been removed at the initial heating.

(6) At some time, the H.P.-phase/L.P.-phase ratio should start to decrease because the sample enters the L.P.-phase stability field. Therefore, we bracket the phase boundary with the lowest pressure where the H.P.-phase/L.P.-phase ratio increases and the highest pressure where this ratio decreases.

(7) We decrease the sample temperature by several hundred K to stop the progression of the transition, compress the sample to a pressure sufficiently above the phase boundary determined in the previous temperature condition, and heat it to a higher temperature by 50 or 100 K than in the previous stage.

(8) We repeat the procedure from Step (5) to (7) to bracket the phase boundary every 50 or 100 K until the phase transition does not proceed due to the long time annealing.

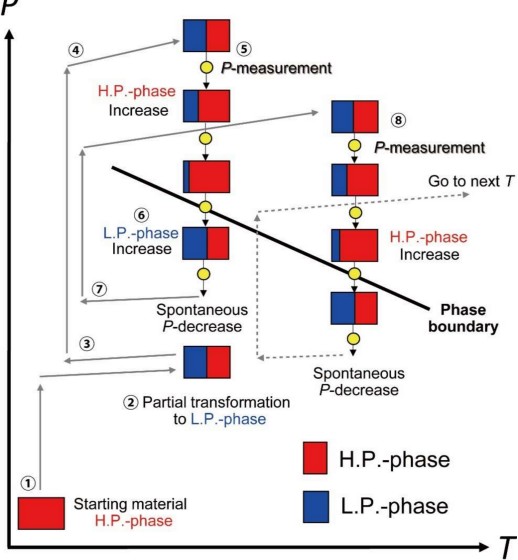

**Figure 4.** A schematic drawing explaining the novel strategy to determine a phase boundary accurately and precisely. The pressure spontaneously and gradually decreases while keeping the temperature and press load constant. In the H.P.- and L.P.-phase stability fields, the H.P.- and L.P.-phases increase with time, respectively, enabling the bracket of the phase boundary. The numbers in the circles indicate the steps in the novel strategy explained in Section 4.

The remarkable point of this procedure is that the phase boundary is bracketed while sample pressure spontaneously, slowly, and monotonically decreases at the fixed temperature. Because the pressure decrease during this procedure is slower than the pressure measurement, and temperature is constant, the obtained pressure–temperature conditions for bracketing a transition pressure are unambiguous.

## 5. A Practical Example to Determine a Phase Boundary: The Post-Spinel Transition in $Mg_2SiO_4$

Here, we give an example of the procedure explained above using the phase boundary of the post-spinel transition by Chanyshev et al. [17]. The L.P.- and H.P.-phases, respectively, in the above explanation correspond to ringwoodite and bridgmanite + periclase.

### 5.1. Preparation of Highly Reactive Starting Material

The starting material of the in situ experiment was a bridgmanite + periclase aggregate synthesized from forsterite powder at a pressure of 27 GPa and a temperature of 1200 K. Since the stability field of bridgmanite is 23–110 GPa [18,29], and the ultrahigh-pressure multi-anvil technology can generate pressure over 50 GPa [30–32], we can synthesize bridgmanite + periclase at 50 GPa and probably at a much lower temperature than 1200 K because of larger excess Gibbs energy caused by significant excess pressure. However, the product amount in 50 GPa multi-anvil experiments is limited, whereas we need a much larger amount of the starting material from a practical viewpoint, we limited the synthesis pressure to 27 GPa. It is also noted that we should minimize the synthesis duration to keep the high reactivity. Our experience of in situ X-ray diffraction of mantle minerals suggests that 5 min is sufficient to convert forsterite to bridgmanite + periclase. Therefore, the heating duration was also 5 min for synthesizing the current bridgmanite + periclase mixture. Note that synthesizing akimotoite from enstatite requires a slightly higher temperature of 1300 K and a much longer duration of 60 min [17].

Figure 5 shows a diffraction pattern of the synthesized bridgmanite + periclase starting material taken using a microfocused X-ray diffractometer. Although the bridgmanite peaks are not very sharp, they can be clearly identified. The lower pressure of 26 GPa produced not bridgmanite + periclase but ringwoodite + akimotoite + periclase. The presence of akimotoite might be the metastable occurrence, as suggested by Kubo et al. [33], in which akimotoite and periclase appeared as alternating lamellae of these phases as an intermediate step of the post-spinel transition. Even at 27 GPa, the product was ringwoodite + akimotoite + periclase when the duration was only 1 min. Thus, the duration of 5 min is necessary, as our experience suggested.

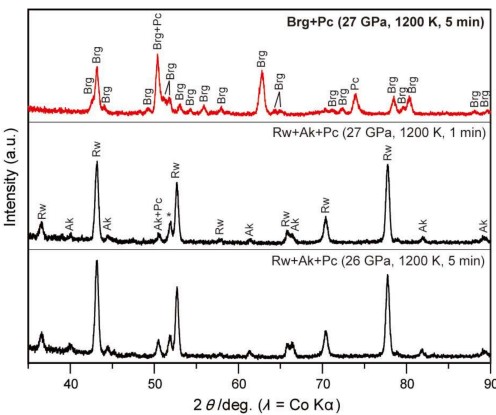

**Figure 5.** Preparation of the starting material of bridgmanite (Brg) + periclase (Pc). After trial runs changing pressure, temperature, and time, the synthesis condition was determined to be 27 GPa and 1200 K for 5 min (**top**). Akimotoite (Ak) and ringwoodite (Rw) appear when the synthesis pressure and heating time is lower and shorter, respectively.

*5.2. Determination of Phase Stability by X-ray Diffraction Profile*

We loaded the starting sample of a bridgmanite + periclase aggregate in the multi-anvil cell assembly shown in Ishii et al. [16]. Its diffraction pattern before compression is shown in Figure 6a. It was first compressed to 26.3 GPa at room temperature, where the press load was 6 MN. Although the sample diffraction should have become diffused if the sample had been powder, the diffraction lines remained sharp due to the aggregate form (Figure 6b). The sample was then heated to 1100 K, where no phase transition was expected. However, we actually observed small amounts of ringwoodite (~35.7 keV: $d$ = ~2.77 Å) (Figure 6c), probably due to few stress-induced defects formed in cold compression. Note that the temperatures given in this article were simply indicated by the thermocouple, whereas those in Chanyshev et al. [17] were pressure-corrected by Nishihara et al. [34]. Sample pressure considerably dropped to 18.9 GPa by heating after cold compression due to the release of deviatoric stresses (Figure 1). Once the sample experiences such a sudden pressure drop, sample pressure does not decrease rapidly and significantly unless the temperature is not significantly increased.

Then, the sample pressure was increased to 21.8 GPa at the same temperature, when the press load was 7.7 MN. The sample was kept for ~1 h at the conditions to find that pressure dropped only from 21.8 to 21.5 GPa, and the sample essentially remained unchanged (Figure 6d). Note that the peak of ringwoodite did not grow after the peak first appeared. Thus, we did not determine the transition pressure at 1100 K.

Then, the temperature was increased to 1200 K at the same press load, which was the starting temperature to determine a phase transition pressure. The pressure was increased to 21.7 GPa by the heating to produce ringwoodite (Figure 6e). The sample was kept at this temperature and press load until the fractions of bridgmanite + periclase and ringwoodite became nearly equal based on the peak intensities (Figure 6f).

Then, we decreased the temperature to 1000 K, increased the press load to 10.5 MN, and increased the temperature again to 1200 K, where the pressure was 24.1 GPa and was expected to be above the phase boundary. We kept the sample at this temperature and press load. We alternatively acquired X-ray diffraction patterns of the pressure marker and sample. Since the reaction was very limited at this temperature, we set the pressure to 24.1 GPa, which is 0.5 GPa higher than the expected transition pressure at 1200 K, to grow peaks of bridgmanite + periclase (Figure 6g).

The sample pressure slowly dropped spontaneously with time, even with keeping temperature and press load, as mentioned in the previous section. However, the pressure drop ceased at 23.8 GPa despite no observation of a decrease in bridgmanite + periclase to ringwoodite ratio.

Hence, we gradually decreased press load from 10.5 to 9.2 MN while monitoring pressure and intensities of ringwoodite and bridgmanite + periclase at 0.5 MN step. We finally observed an intensity change at 23.1 GPa while the press load of 9.2 MN was kept for 30 min (Figure 6h). This was by ~0.5 GPa lower than the transition pressure estimated from those determined at higher temperatures [17]. Thus, the transition pressure was constrained with a pressure interval of ±0.5 GPa, which is about 10 times larger than precision of pressure measurement, indicating that we did not sufficiently constrain the transition pressure at 1200 K due to the sluggish kinetics.

Because of no further reaction, we decreased the sample temperature by 200 K and increased the press load to 11 MN, when the pressure was 23.8 GPa, above the phase boundary. Then, we increased the temperature to 1250 K, 50 K higher than the previous step, and determined the transition pressure by repeating the same procedure. Although we could not observe the increase in bridgmanite + periclase/ringwoodite ratio, probably due to insufficient excess pressure, we observed the decrease in this ratio.

With increasing temperature, the reaction rate gradually increased, and the transition pressures were constrained more tightly. The pressure interval was ±0.3 GPa at 1300 K, and ±0.15 GPa at 1500 K. At 1550–1650 K, the transition pressures were very tightly constrained within a pressure interval of ~0.1 GPa.

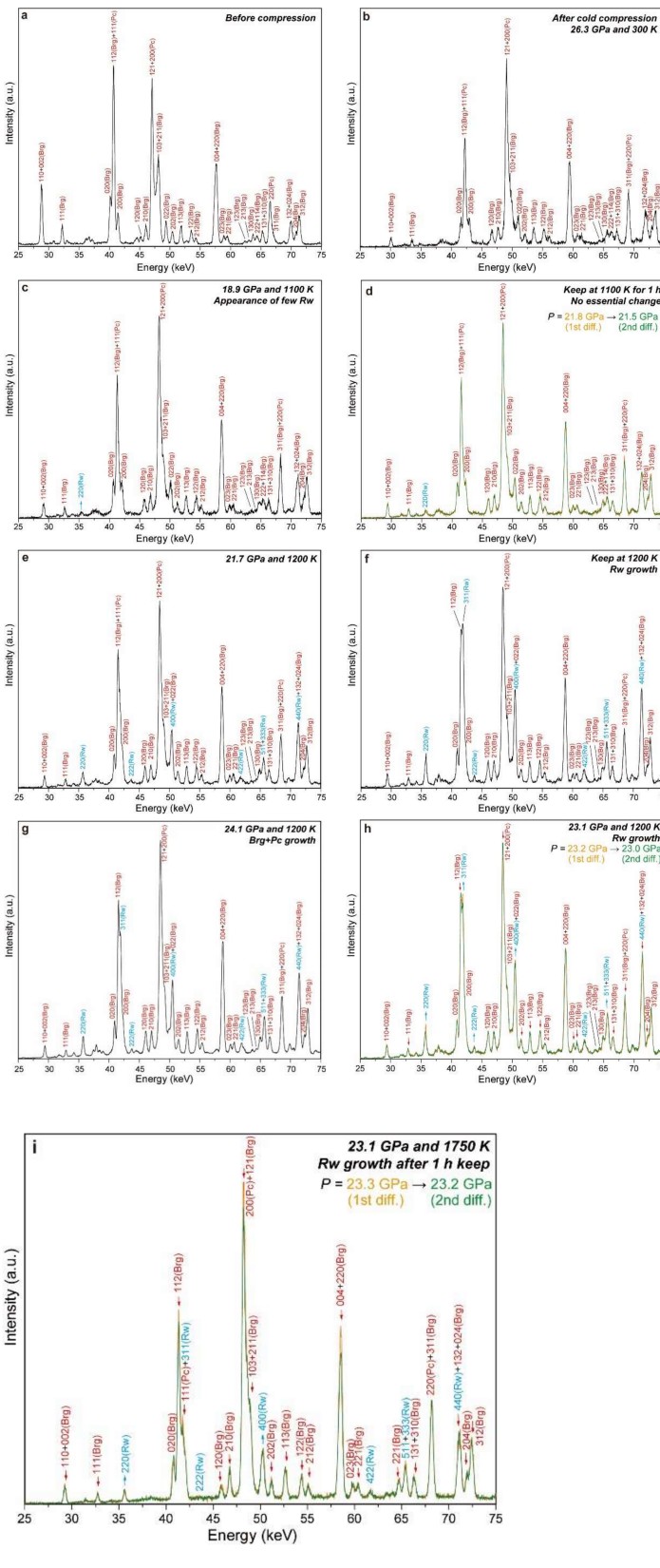

**Figure 6.** Change in X-ray diffraction patterns of the sample: (**a**) before compression; (**b**) after cold compression; (**c**) after reaching 1100 K; (**d**) maintaining at 1100 K for 1 h; (**e**) after heating to 1200 K; (**f**) after maintaining at 1200 K; (**g**) after increasing pressure to 24.1 GPa at 1200 K; (**h**) during spontaneous pressure decrease from 23.2 to 23.0 GPa at 1200 K; (**i**) during spontaneous pressure decrease from 23.3 to 23.2 GPa at 1750 K.

However, the reaction rate gradually became low with increasing temperature due to the more potent effect on sample reactivity of the grain growth and lower point- and line-defect densities than temperature increase, as explained. The constraint of the transition pressures became $\pm 0.2$–0.3 GPa at most temperatures above 1700 K. The change in the intensity ratio became subtle even though we waited hours to detect it (Figure 6i). We terminated the determination of this transition at a temperature of 1950 K, due to the limited synchrotron beam time despite the required long reaction times.

During this procedure, the sample assemblage may be completely converted to the H.P.-phase or the L.P.-phase due to the balance of fast transformation kinetics with increasing temperature and degree of excess pressure. Fortunately, both phases always survived during our experiment, probably because we gradually increased the temperature from 1100 to 1950 K by spending more than 2 days. As a result, the sample became very inert even at 1950 K, determining phase stability by slight intensity change. Once either of the phases disappeared, growing the disappeared phase was not easy because nucleation of new grains needed excess pressure, as mentioned above. According to the kinetic data by Kubo et al. [34], the time for the 10% transformation of ringwoodite at 1200 K is about 76 days at an excess pressure of $1.0 \pm 0.5$ GPa, whereas about 19 min at an excess pressure of $4.5 \pm 1$ GPa. For the post-spinel transition in $Mg_2SiO_4$, excess pressures of 3.5–5.5 GPa thus could be necessary to grow bridgmanite + periclase again. The growth of ringwoodite would need a similar pressure decrease from the phase boundary.

## 6. Discussion

### 6.1. Comparison with Previously Determined Phase Boundaries

The post-spinel phase boundary obtained has a very gentle concave curve (Figure 7). The post-spinel phase boundary is located at pressures of 23.2–23.7 GPa in the temperature range 1250–2040 K. Its slope changes from $-0.1$ MPa/K below 1700 K to $-0.9$ MPa/K above 1700 K. Previous studies reported relatively large negative slopes of –3 to –1.3 MPa/K [11,13,18,22] or a relatively small slope of ~–0.4 MPa/K [19,23]. Previous studies determined by the quench method show relatively large negative slopes of –2 to –2.5 MP/K [11,13]. Fei et al. [18] constrained the phase boundary based on quench experiments in combination with in situ measurement of pressure. Their slope is a smaller value than those by these quench experiments but also a relatively large negative value of –1.3 MPa/K. These studies observed the formation of high-pressure phases from a starting material of powdered forsterite. Wider stability of ringwoodite at relatively low temperatures can be explained by metastable ringwoodite initially formed during heating, showing larger negative slopes. Irifune et al. [22] also showed a large negative slope of ~–3 MPa/K based on continuous in situ X-ray diffraction experiments. They mainly determined the phase boundary by detection of ringwoodite from bridgmanite + periclase during decompression. At higher temperatures, the samples were annealed to grow the grains and decrease the point and line defects, requiring larger negative excess pressures and leading to the steeper negative phase boundary. Katsura et al. [23] and Ghosh et al. [19] reported relatively small negative values of ~–0.4 MPa/K as the most probable value by continuous in situ X-ray diffraction experiments, which is close to our result. They carefully considered kinetic effects of phase change, which could have avoided misinterpretation of stable phase. On the other hand, due to necessary excess pressure, their slope values were largely scattered (Katsura et al. [23]: +1.2 to –2 MPa/K and Ghosh et al. [19]: +0.63 to –0.69 MPa/K). One reason for the various excess pressure by Katsura et al. [23] is because they induced the post-spinel transition by rapid temperature increase, causing complex pressure increase and decrease as explained in Section 3. Ghosh et al. [19] reported the steep negative slope in the wet system with 2 wt.% $H_2O$. However, the slope is invalid because they did not observe the growth of ringwoodite from bridgmanite + periclase. Thus, our new method only reaches a conclusion of a small negative slope.

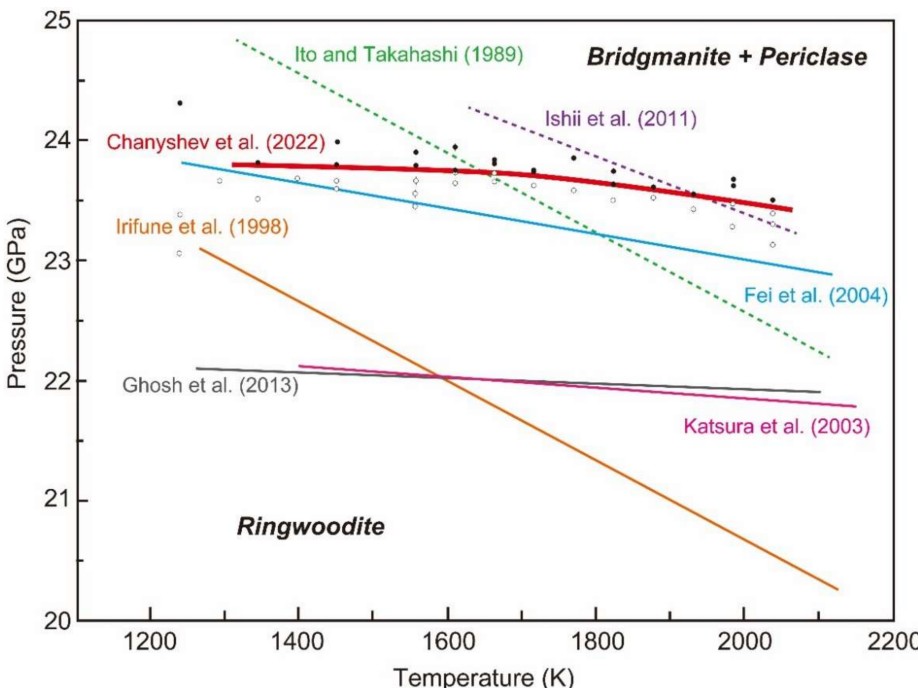

**Figure 7.** Comparison of the post-spinel phase boundary in $Mg_2SiO_4$ obtained by the novel technique with previously obtained boundaries. Dashed boundaries were determined by quench method. Solid lines were determined by in situ X-ray diffraction technique. Open and closed circles indicate stable pressure–temperature conditions for ringwoodite and bridgmanite + periclase, respectively [11,13,17,18,22,23].

*6.2. Mantle Phase Transitions to Be Redetermined Using the New Strategy*

The determinations of many mantle phase transitions did not follow the principle of phase equilibrium and should have suffered from the pressure change upon heating, lower-temperature phase transitions at threshold temperatures, and sluggish kinetics. Therefore, such phase transitions should be redetermined using the current strategy. On the other hand, the current strategy is applicable to univariant transitions but not to multivariant transitions. For these reasons, we should reinvestigate the following univariant coexistences: pyrope + bridgmanite + corundum in $MgSiO_3$-$Al_2O_3$ [20,35], wadsleyite + bridgmanite + periclase in $Mg_2SiO_4$ [13,28], ringwoodite + stishovite + wüstite + bridgmanite + iron in $MgO$-$SiO_2$-$Fe$-$O$ [11,13], stishovite + wüstite + bridgmanite + iron in $MgO$-$SiO_2$-$Fe$-$O$ [11,13], majorite + bridgmanite in $MgSiO_3$ [21,36], majorite + akimotoite in $MgSiO_3$ [13,36], wadsleyite + ringwoodite in $Mg_2SiO_4$ [12,37,38], high-pressure clinoenstatite + majorite in $MgSiO_3$ [36], orthoenstatite + high-pressure clinoenstatite in $MgSiO_3$ [39,40], forsterite + wadsleyite in $Mg_2SiO_4$ [12,24,41], forsterite + wadsleyite + ringwoodite in $Mg_2SiO_4$-$Fe_2SiO_4$ [12], coesite + stishovite in $SiO_2$ [42,43], stishovite + wüstite +ahrensite + iron in $SiO_2$-$Fe$-$O$ [44], and ahrensite + fayalite in $Fe_2SiO_4$ [27].

**Author Contributions:** Conceptualization, T.I. and T.K.; methodology, T.I. and T.K.; investigation, T.I., A.C., and T.K.; writing—original draft preparation, T.I.; writing—review and editing, A.C. and T.K.; funding acquisition, T.K. All authors have read and agreed to the published version of the manuscript.

**Funding:** This research was funded by the European Research Council (ERC) under the European Union's Horizon 2020 research and innovation programme (Grant agreement No. 787 527).

**Data Availability Statement:** All data are shown in this article.

**Acknowledgments:** T.K. thanks Julien Chantel for invitation to the Special Issue: In Situ Measurements of Physical Properties of Rocks, Minerals, and Fluids at Extreme Conditions. We thank three anonymous reviewers for their constructive comments.

**Conflicts of Interest:** The authors declare no conflict of interest.

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
