# Peer review of "A New Approach Determining a Phase Transition Boundary Strictly Following a Definition of Phase Equilibrium: An Example of the Post-Spinel Transition in Mg2SiO4 System"

_minerals, doi:10.3390/min12070820_

Round 1

Reviewer 1 Report

Phase transitions of minerals are very important to understand the mantle dynamics and evolution. In this manuscript, the authors expound in detail the method to investigate the phase boundary of minerals under high pressure and temperature based on the definition of phase equilibrium, although the major ideas and experimental results have been published in Nature (2022). These informations are very important for experimenters. The following contents need to be explained and revised.

1 In figure 2, the authors defined the average pressure of the sample, , where D0 is the time of an initial pressure first rapidly increase in few seconds due to thermal pressure. However, in line 102-104, the authors pointed out “This pressure increase should happen instantaneously, although our experimental technique cannot observe it…..”. It indicates D0 is very difficult to determine in the experiments. Thus the parameter of Pm is very difficult to evaluate accurately. And how to accurately determine the parameter D1? Whether it is significant to define Pm?

2 In figure 3, in general, what is the magnitude of the excess P?

3 In lines 204~208: if the L.P phase transforms to H.P phase during cooling temperature and increasing pressure process in step 3, what should to be done on the next step?

4. In lines 209~215: In the step 5, during the pressure spontaneously drops, H.P phase would transform to L.P phase at constant temperature. However, the authors wrote “L.P.-phase should slowly transform into H.P. phase at the beginning… ” in line 213. Whether these contents are contradictory?

5 In lines 213~215If the pressure decreasing is artificially instead of spontaneously, does it affect the data of Pm? Why the pressure artificially decrease during the process?

6 How to judge the pressure drop in a muti-anvil pressure is due to the phase transition or material softening in the experiments?

6 Some figures are missing in figure 6.

7 what does MN mean?

Reviewer 2 Report

I have made most of my individual comments in the PDF file of the manuscript, which I have attached here. However, there are a couple of major points that need to be addressed, and I will discuss them here.

1) Figure 6 - The entirety of Section 5.2 is based around Figure 6, and it did not transmit into the PDF properly. It goes g, h, i, d, e, f instead of starting with a, b, and c. There are double axes awkwardly placed, and I can't interpret the discussion in 5.2 properly because of the problems with this figure. I suspect the previous panels are somehow underneath the ones shown here, which is in part what causes the double axes, but I cannot view them no matter what I try. Therefore, I have been unable to properly assess section 5.2, and since the conclusions are based on this, I can't fully assess them at this time either.

2) No information is given about the in situ experiments themselves. What beamline was used and where? What are the characteristics of the beam? Types of detectors used? All of this information plays a part in understanding whether the procedures presented here are novel and valid, and it is unfortunately absent. There is even no mention in the acknowledgments. This is a major piece of the puzzle and needs to be included.

Overall, I think the work presented is well thought out and meticulously conducted, but based on the two major items discussed above, I cannot fully assess its suitability for publication. I am happy to revisit a revised version of this manuscript when it is ready.

Reviewer 3 Report

Chayshev et al. report a new determination of the P-T phase boundary line of the post-spinel transformation in Mg2SiO­­4 using data from a previous report from the same group [Nature 601, 69-73, 2022]. The new method aims to achieve a better constraint of the P-T phase boundary line by addressing the problems that arise with instantaneous pressure increases due to thermal pressure, pressure drops during heating, and sluggish reaction kinetics. This paper does a good job of laying out potential issues with previous studies, and devotes a good amount of effort to a pedagogical walkthrough of the new methodology used. However, all of the data reported In this paper were already reported in the Nature paper, and it might have been more appropriate to provide this discussion in that paper, perhaps as supporting information.

Issues to address:

Figure 6 is mislabeled, making it very difficult to follow along with the main text. The main text refers to subfigures A–F, but they are labeled g, h, i, d, e, f.

The described experimental steps do not match the schematic in Figure 4. For example, line 292 appears to be describing Step 5, which according to Figure 4 should be the synthesis of the HP-phase, bridgamanite+periclase. However, the text describes the synthesis of the LP ringwoodite. Section 5.1 would be greatly improved by referring to the specific steps outlined in Figure 4, to help the reader follow along.

It would be useful to plot the data used to determine the phase boundary that is then presented in Figure 7. This would help the reader to assess how tightly bracketed the proposed slope is based on the method presented. The authors could recreate Figure 2a from the Nature paper, since they are reusing all of the data from that paper anyway.

Section 6.2 is not an efficient use of space. These systems could be listed in sentence form.

In summary, this is a well-written paper that provides a detailed account of a novel technique for determining more reliable phase boundaries. However, the lack of novel data diminishes the value of this paper somewhat.
